# Wildfire impacts and mitigation strategies among California cannabis producers

Jeff Vance Martin[1,2¤a]*, Christopher Dillis[1,2], Genoa Starrs[1,2¤b], Danielle Schell[1], Theodore E. Grantham[1,2], Van Butsic[1,2]

1 Berkeley Cannabis Research Center, University of California, Berkeley, California, United States of America, 2 Department of Environmental Science, Policy, and Management, University of California, Berkeley, California, United States of America

¤a Current address: USDA Forest Service, Pacific Southwest Research Station, 1700 Bayview Street, Arcata, California, United States of America
¤b Current address: University of California, Agriculture and Natural Resources, Informatics and GIS Statewide Program, 125 Mulford Hall, Berkeley, California, United States of America
* j.vance.martin@gmail.com, Jeffrey.Martin4@usda.gov

## Abstract

California has experienced increasing frequency and intensity of wildfire, with the five largest fires on record since 2018. Over the same period, licensed cannabis production has grown to a high-grossing industry, while remaining an important source of rural livelihood. Importantly, the geography of cannabis production overlaps with high fire hazard areas more than any other crop in the state. We developed and deployed a state-wide survey of licensed outdoor producers to determine direct and indirect impacts of wildfire, as well as how producers have responded to these threats. Quantitative and narrative data were subjected to statistical and thematic analyses, demonstrating key findings around fire-related losses, mitigation tools and techniques, and perceptions of risk. Producers experienced a range of impacts beyond direct burning, including reduced light (affecting grow rates), ash deposition (with impacts on product quality and saleability), and production disruptions. Producer responses to the threat of fire and smoke varied, in part affected by the costs of mitigation, yet some common strategies emerged. However, while most growers reported impacts from fire, these were often outweighed by concerns over other pressures on production and profitability. Our hope is that these findings around the experiences and concerns of California's cannabis producers will inform future research directions and provide the first steps toward policy interventions to better address the challenges of living with wildfire.

## Introduction

California is experiencing an increasing frequency and intensity of wildfire, bound up with a longer history of land management and climate change. While fire had been foundational to the California landscape through historical patterns of natural ignition and indigenous burning, over a century of fire suppression – alongside increasing development in the wildland-urban interface and growing aridity – has made wildfires in the state increasingly frequent, intense, and destructive [1–5]. The five largest California wildfires on record have occurred since 2018, with 2020 the largest fire year in the state's recorded history thus far [6,7].

**Data availability statement:** Data cannot be shared publicly because of ethical restrictions: data contains potentially identifying information in qualitative responses and in provision of geographic location data. Sharing data would represent a breach of compliance with our human subjects ethics approval (categorization: Exempt), as negotiated with the Committee for Protection of Human Subjects (CPHS), University of California, Berkeley (CPHS Protocol: 2022-02-15030). For further information, please see S2_IRB_Approval_ Letter (attached). Fellow researchers may contact the corresponding author (JVM) for data inquires, and/or the University of California, Berkeley's Office for Protection of Human subjects at ophs@berkeley.edu.

**Funding:** This work was supported by the California Department of Cannabis Control (https://cannabis.ca.gov/), grant number 65303 (grant awarded to VB). The funders played no role in study design, data collection and analysis, decision to publish, nor preparation of the manuscript.

**Competing interests:** The authors have declared that no competing interests exist.

Over this same period, California's licensed cannabis industry – established in 2018 – has also grown to become one of the top five grossing agricultural products in the state [8].

Much of California wildfire takes place in rural or semi-rural landscapes, including at the wildland-urban interface, or WUI [9,10]. Despite ongoing transformations within the cannabis industry, legacy producers in rural Northern California retain the greatest number of outdoor cannabis farms and represent a large proportion of total state production [11], and cannabis remains a major driver of rural economies [12]. As previously documented, cannabis' existing geography of production overlaps with areas of high fire hazard more than any other agricultural type in the state [13]. Where wildfire and smoke impacts overlap with California's outdoor cannabis farms, they put the economic and physical well-being of rural producers and their employees at risk [8,14]. Such concerns over the human impacts of disaster build from a political ecology-informed understanding of environmental hazards, vulnerability, and risk perception as geographically situated, historically constituted, and socio-environmentally co-produced [15–19].

Although there is widespread anecdotal evidence of damage to cannabis production from the direct and indirect effects of wildfire (i.e., through burning, smoke exposure, and ash deposition), there has been little to no research identifying the relative occurrence and importance of these impacts and/or the mitigation strategies in use by cannabis producers across the state. This project, begun in 2021 through support from the California Department of Cannabis Control (grant number 65303), aims to begin filling these gaps with the hope of informing future research and policy (see also [20]). Earlier publications from this work have focused on the cannabis industry's particular vulnerability to wildfire [13], and the economic impacts of smoke exposure [8]. Here we focus on the broader array of wildfire-related impacts and producer responses. Based on our statewide survey of licensed cannabis producers, we ask: 1) *What are the direct and indirect impacts of wildfire on outdoor cannabis production?* and 2) *How are cannabis producers responding to the threats of wildfire and smoke?*

In what follows, we provide background on the patterns and history of California wildfire and the licensed cannabis industry as read through a political ecology lens, before turning to our materials and methods. This is followed by our findings, exploring in detail each of the research questions above. We then turn to an analytical discussion that situates wildfire mitigation within a broader socio-environmental context, before concluding with a few suggestions for future research. It is our hope that this work provides new insights on the experiences and adaptive strategies of cannabis producers contending with California's "new normal" of wildfire, as well as guidance for on-the-ground mitigation and state- and county-level policy.

## Background and methods

### Wildfire and cannabis in California

California's landscape has been fundamentally shaped by fire. Indigenous nations conducted landscape-scale fire stewardship for thousands of years to promote desired habitats and species [21,22], alongside natural ignitions from lightning strikes. However, a history of fire suppression beginning in the early 1900s, combined with the expansion of development in the WUI and growing drought and aridity bound up with climate change, has resulted in patterns of increasing intensity and severity of wildfire in California and across much of North America [1,3,5,23–27]. Increasingly described as a "wildfire crisis," this new fire regime has resulted in growing rates of property destruction, ecosystemic devastation, and loss of life [28–30].

Scholars in political ecology and cognate fields have long challenged the concept of the "natural disaster" by instead highlighting the socio-political co-production of environmental

hazards and uneven experience of vulnerability [18,31]; see also [15–17]. As much as wildfire is a force that exceeds human intention and control [32], it is bound up with histories of human-driven land use and climate change, and unevenly experienced alongside other pressures by situated social actors (compare [33,34]). The conditions that give rise to wildfire in particular times and places – e.g., the buildup of fuel loads as a consequence of fire suppression [5,35] – and the relative vulnerability of particular populations and regions bound up with broader patterns of inequality and policy [16,19,30], are products of human doing. Indeed, to contend that wildfires are simply *natural* risks obscuring and depoliticizing their origins – shot through with relations of power – and undermining the possibility of conditions and management patterns otherwise [36].

Drawing on these insights, we conceptualize fire as co-produced through the interaction and interrelation of natural and social factors – a framing that informs our study design and analysis (note also the previously argued value of applying political ecology insights to western landscapes – see [37–39]). A political ecology orientation thus highlights "how biophysical and social processes shape local conditions and the adaptive capacity of particular populations and places" [19]. The geography of cannabis production has likewise been socio-ecologically produced. A history of il/legality, combined with a California landscape shaped by agriculture and fire suppression, has resulted in disproportionate overlap between cannabis production and fire-prone environments [13,40] – even as the physicality of smoke connects local vulnerabilities to physically distant events [8]; see also [41]. Crucially, questions of adaptation and mitigation also hinge on context and positionality at different scales, affected by uneven patterns of regulation, economics, and geography.

Rural California and much of the Pacific Northwest has long been characterized by a history of resource-based boom and bust economies. The decline of timber harvests beginning in the 1990s had notable impacts on community well-being across the region [42–44], with subsequently varying development trajectories, including patterns of "New West" rural gentrification and recreation-oriented economies [45–48]. Some communities sought alternative revenue streams, including via cannabis cultivation – long before state-level legalization – particularly in the so-called "Emerald Triangle" of Humboldt, Trinity, and Mendocino counties in Northern California.

Since the passage of Proposition 64 in 2016, California has developed administrative systems under the Department of Cannabis Control (DCC) for licensing and regulating the cultivation, distribution, and sale of cannabis for recreational and medicinal use, including testing of all products for residual pesticides and other potentially harmful chemicals. While significant changes continue to unfold – in the regulation, production patterns, and geography of California's cannabis industry [11,49,50] – rural production in "legacy" growing regions remains an important source of local livelihoods [12,51], while being among the worst affected areas in California's new regime of devastating wildfires. (A quick note on terminology: we refer to "producers" throughout this piece, but both "farmer" and "grower" were used more or less interchangeably by informants. Our research did not delve deeply into the differing positionality and ownership/labor structures of cannabis production, although we address some of these concerns in the discussion.)

## Survey design and analysis

Survey development was based on earlier research and responses from focus groups with cannabis producers and other industry-associated actors. Prior work [11,13,50] demonstrated geospatial overlap between areas of cannabis production within California and zones of high fire risk, in part related to a history of illegality (as noted above). Focus group discussions were

conducted by the lead author online between late 2021 and early 2022 with producers and other industry-associated actors (n = 12) recruited through known contacts of the Cannabis Research Center (CRC) and via a snowball sampling approach [52]. These guided conversations covered impact types and perceptions, mitigation practices, and related challenges and concerns, and were used to inform subsequent survey design.

Our survey was designed around two broad research questions, as noted above, focusing on 1) direct and indirect *impacts* of wildfire on outdoor cannabis production, and 2) producer *responses* to the threats of fire and smoke. Data collected covered *operations characteristics*, including respondent role, farm location and size, length of time in production, industry associations, production forms and product type, and license status. We asked producers to report on *wildfire impacts*, including experiences of direct fire losses, smoke exposure, ash/particulate accumulation, effects on product (including quality impacts and economic losses), testing patterns and results, and other fire-related impacts on their crops and businesses (e.g., health and safety concerns and disruption of production and distribution). We also queried producer adaptations and *mitigation strategies* in use, including insurance status, installation and use of various technologies and techniques, their perceived effectiveness, costs, and overall changes in production practices (including timing and location). The survey also included questions regarding *attitudes* toward various governmental agencies as well as broader sources of concern among producers (see Survey Protocol for additional details – S1 File).

Survey questions covered the 2018–2021 growing seasons – although not all respondents cultivated cannabis in each year – and focused on outdoor and mixed-light licenses. We chose this timeframe because it represents the first years of cultivation under Prop 64 and aligns with the most active wildfire years in California's recent history. Full-sun outdoor and mixed-light farms represent the majority of cannabis production in California. Mixed light production uses light deprivation to simulate rapid seasonal progression, allowing for multiple grow cycles per year; although artificial lighting may be used, it is not to the same extent as fully indoor production, nor is mixed-light fully isolated from the outside environment. According to the DCC, mixed-light licenses are for producers using greenhouses, hoop-houses, glasshouses, conservatories, hothouses, or other similar structures [53]. We focused on these growing techniques as we expected them to be more vulnerable to wildfire and smoke than fully indoor farms, which generally employ closed and filtered HVAC (Heating, Ventilation, and Air Conditioning) and are often situated in more urban areas.

Questions included a mix of multiple choice and short answer, along with opportunities for additional elaboration through fill-in-the-blank "Other" options on many questions and space at the end of the survey for respondents to include additional information. These narrative responses provided nuance and detail that helped us to better interpret the patterns observed in our quantitative findings and were approached both by individual question and as a whole (see below).

Our survey instrument received IRB approval from the Committee for Protection of Human Subjects (CPHS), University of California, Berkeley (CPHS Protocol Number 2-022-02-15030 (Exempt)). The survey included a dialog box clarifying data security, anonymity, and requesting (unsigned) consent. Survey respondents were recruited via email using addresses obtained from the DCC (license data for California outdoor and mixed-light cannabis farms for the years 2020–2021) [54]. Data were collected anonymously using the Qualtrics survey platform (Qualtrics, Provo, UT), with promotion conducted through Qualtrics follow-up messages and via social media, industry associations, and the website of the CRC (crc.berkeley.edu).

Survey responses (n = 199, representing an estimated response rate of 13%) were analyzed using R Statistical Software (R Core Team, 2018). Descriptive statistics were used to

summarize categorical survey responses, and responses are reported in aggregate across all study years. Narrative responses were subjected to thematic analysis through an inductive, interpretive approach that identified prominent themes and patterns within the open-ended answers we received [55], representing an instance of data triangulation between quantitative and qualitative data sources [52,56] (on the value of qualitative data and methods, see [57,58]).

## Results

### Direct and indirect impacts of wildfire on outdoor cannabis

Survey responses were most common from Northern Californian counties, with the top three response counts from Humboldt (n = 41, 29% of total), Trinity (n = 32, 23% of total), and Mendocino (n = 30, 21% of total) counties. Although much of California cannabis production now occurs outside the Emerald Triangle, this is largely due to differences in farm *size*; the largest *number* of individual farms are still in Northern California [11], making our sample proportional to geographical distribution (rather than production totals). There is likely a self-selection bias present among our respondents toward those who have experienced (or are concerned about experiencing) impacts from wildfire. However, given that impacts are more associated with smoke exposure than proximity to fire, and given the distance that smoke plumes can travel [8], we maintain that these concerns are a statewide issue.

The vast majority (86%) of our respondents reported human health impacts from exposure to wildfire smoke (Fig 1). A decline in crop growth rate was reported by 70% of respondents, explained by many as a result of natural light reduction due to smoke. Smoke plumes can block sun exposure during key points of the growing season, with reported effects on maturation rate, yields, and potency.

Reported crop impacts can be compared to reported smoke and ash levels. Table 1 shows crop impacts – categorized in our survey as *negligible impact*, *reduced value*, or *unsellable* – along with reported days under smoke (light and thick) and ash accumulation on plants (see Survey Protocol for additional details – S1 File). These data appear to follow expected patterns, demonstrating that time under thick smoke results in greater impacts on crop quality relative to light smoke. On its own, however, ash accumulation begins to result in reduced value under only a moderate percentage of occurrence, as compared with light and thick smoke. This seems to indicate that ash deposition has a qualitatively different impact for

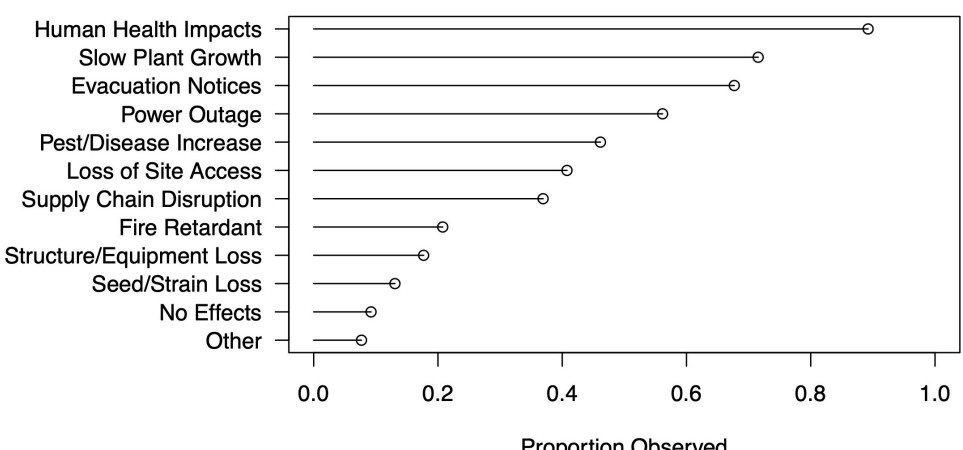

**Fig 1. Wildfire-related effects.**

**Table 1. Reported smoke and ash impacts.**

|  | Negligible Impact | Reduced Value | Unsellable |
|---|---|---|---|
| Light Smoke |  |  |  |
| *Low* | *93%* | *4%* | *3%* |
| *Moderate* | *66%* | *23%* | *12%* |
| *High* | *39%* | *26%* | *35%* |
| Thick Smoke |  |  |  |
| *Low* | *85%* | *12%* | *3%* |
| *Moderate* | *62%* | *17%* | *21%* |
| *High* | *33%* | *28%* | *38%* |
| Ash Fall |  |  |  |
| *Low* | *87%* | *10%* | *4%* |
| *Moderate* | *55%* | *27%* | *19%* |
| *High* | *49%* | *20%* | *31%* |

Reported percent of growing season experiencing light smoke, thick smoke, and ash fall (left; aggregated into classifications of low (0%, 10%), moderate (25%, 50%), and high (75%, 100%)) – compared with reported impacts on crop value (top; percentage of respondents reporting).

cannabis plants, matching anecdotal evidence and producer conversations – some of whom noted that wet ash, in particular, could form a "crust" that resulted in full loss of product. These impacts may also result from a greater frequency of pests or fungi (45%, Fig 1), as producers noted powdery mildew, yeast, and microbial growth following exposure to smoke and particulate matter.

We also examined the cumulative effects of thick smoke and ash deposition (Table 2). Large percentages of respondents reported reduced value and unsellable crops at high levels of thick smoke, even when ash accumulation was low (i.e., infrequent). In comparison, there were more reports of negligible impact when ash accumulation was high (i.e., frequent) yet thick smoke was low. Even with high ash accumulation, the majority of respondents reported negligible impact unless there was also high incidence of thick smoke. These results once again suggest that smoke and ash can have cumulative negative impacts but that they act in different ways. It is possible that it is easier to prevent or physically remove ash accumulation than it is to combat smoke taint (something discussed in producers' narrative responses), or that producers are less willing or able to mitigate ash effects when smoke is also present, perhaps due to risk of health effects.

**Table 2. Smoke and ash combined.**

|  | Negligible Impact | Reduced Value | Unsellable |
|---|---|---|---|
| Thick Smoke High |  |  |  |
| *Ash Low* | *40%* | *30%* | *30%* |
| *Ash Moderate* | *40%* | *20%* | *40%* |
| *Ash High* | *29%* | *29%* | *42%* |
| Ash High |  |  |  |
| *Thick Smoke Low* | *62%* | *29%* | *10%* |
| *Thick Smoke Moderate* | *55%* | *10%* | *36%* |
| *Thick Smoke High* | *29%* | *29%* | *42%* |

Following the same classifications as Table 1, reported impacts are presented under conditions of thick smoke and ash together. Values of ash (percentage of days during the growing season) are varied against high values of thick smoke and vice versa to demonstrate cumulative impacts.

For producers who experienced smoke, *smoke taint* – similar to the undesirable sensory characteristics noted in wine production [59,60] – was repeatedly discussed. Cannabis is particularly susceptible to taint during flowering (generally July through October in California), with smoke affecting the smell of the final product and thus reducing market value and/or "making sales difficult," as one respondent put it. Processing tainted flower into oil was noted as one potential method to recoup these losses, yet market saturation – and pivoting to another commodity chain – may prevent many producers from pursuing this approach.

We asked respondents if they had any lab-based chemical testing of their product conducted following smoke or ash exposure. The DCC requires testing to ensure cannabis is free of contaminants (including residual pesticides, solvents, and processing chemicals, as well as heavy metals, microbial impurities, and mycotoxins) and to accurately label for cannabinoid and terpene content [61]. However, such testing is not currently designed to specifically measure the effects of smoke or ash, nor is testing following wildfire/smoke exposure a legal requirement. Of those who did report testing their product during a cultivation year with smoke – only 66 of our 199 respondents – 53 passed and 13 failed, with the latter attributed most often to the presence of heavy metals (46%). Many producers did not have their product tested given the lack of a legal requirement, with some noting how smoke taint or other reductions in quality made testing superfluous.

In addition to the effects of fire, smoke, and ash on product quality, many producers also noted logistical knock-on effects from wildfire on production processes (Fig 1). 66% experienced emergency evacuation notices, which could result in losses if road blockages prevented return and thus watering and/or harvest of crops (loss of site access, 39%). As one respondent explained:

> "For wildfires- the biggest problem was dealing with local roadblocks and evacuation orders. All agencies wanted us to evacuate. You could leave anytime but there was no coming back … If I would have evacuated, I would have lost my entire crop and income for the year which would have put me out of business and I would have been foreclosed on my home as well … After this experience, I will probably never evacuate. Some firefighters were very helpful, while others clearly had a bias against cannabis farmers and were very unhelpful."

Producers also experienced power outages during or around periods of fire (54%), and such outages were reported to impact various on-farm systems, from artificial lighting (for mixed-light producers) to automatic irrigation and security systems. Additionally, 35% experienced supply chain disruptions (e.g., getting harvested crops to market) as a result of fires, which could exacerbate concerns over reduced value.

## Cannabis producer responses to the threats of wildfire and smoke

In the face of growing wildfire and smoke impacts, many growers have made changes in production as a means of reducing risk and increasing resilience. Survey respondents expressed either agreement or disagreement with how fire and/or smoke motivated changes in their decision making, including shifts in on-farm practices, timing, and location (Fig 2). 72% of our respondents agreed that fire or smoke motivated changing *practices* (40% "Strongly Agree," 32% "Somewhat Agree"). In contrast, 40% agreed (either strongly or somewhat) that fire and/or smoke motivated changes in the *timing* of planting or harvest, while only 28% agreed it impacted choice of farming *location* ("where I grow"), and only 16% agreeing that fire/smoke had "motivated a shift toward indoor production."

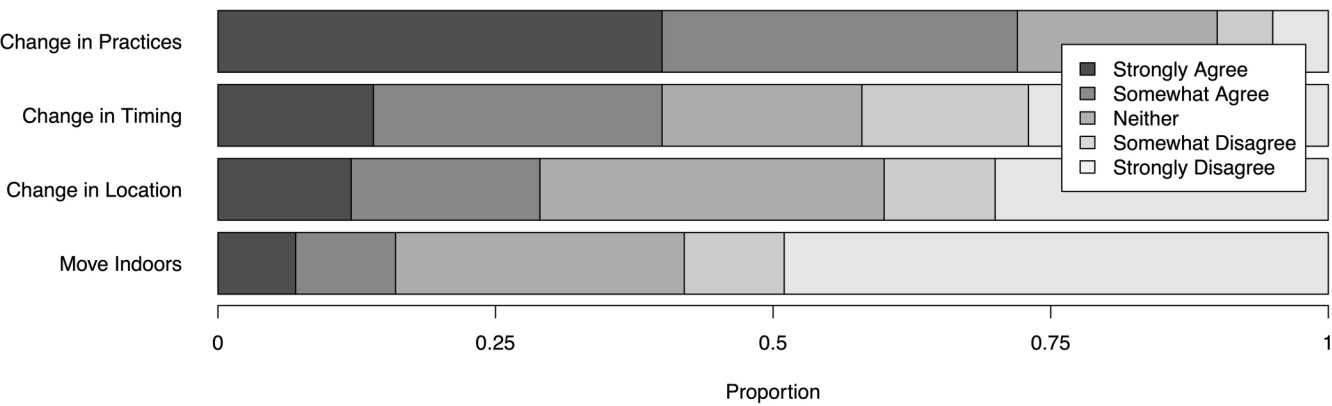

**Fig 2. Changes in farming due to wildfire.**

We asked about a range of practices and technologies – identified in prior discussions and focus groups with producers – to determine the frequency with which such strategies were used by outdoor producers across the state (Fig 3). Respondents overwhelmingly noted usage of fuel reduction (i.e., brush clearing) (81%) and on-site water storage (80%). Also reported in relatively high numbers were provision of personal protective equipment (PPE) to workers (68%), automatic irrigation systems (65%), and fire safety and ignition reduction measures (65%) (the latter often linked in discussions with fuel reduction practices). Washing particulates off growing plants, along with protective infrastructure (e.g., hoop houses) to reduce the impacts of ash deposition, were also commonly reported (57% and 39%, respectively), as were roadway improvements to facilitate site access (i.e., by firefighters) (58%). One notable limitation of our data, however, is that these techniques were not coded to year – thus we cannot say whether usage was adopted following fire/smoke impacts, or already in use unrelated to increased risks.

We asked producers which of the strategies previously noted was seen as most effective for reducing fire/smoke impacts (Fig 4). Interestingly, while fuel reduction was both widely

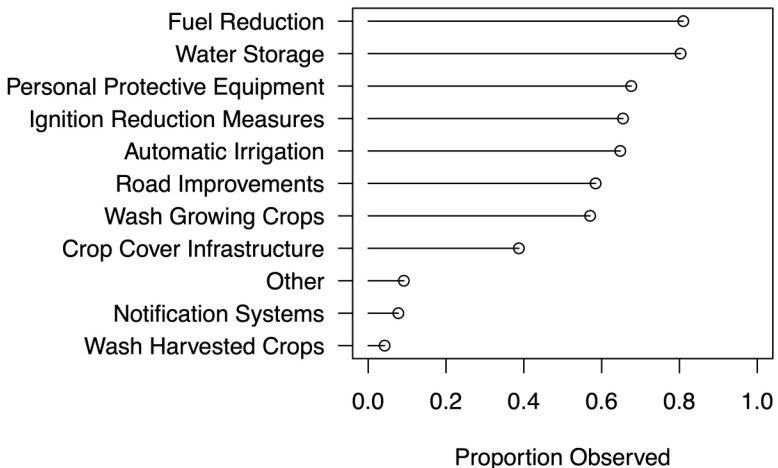

**Fig 3. Mitigation strategies reported.**

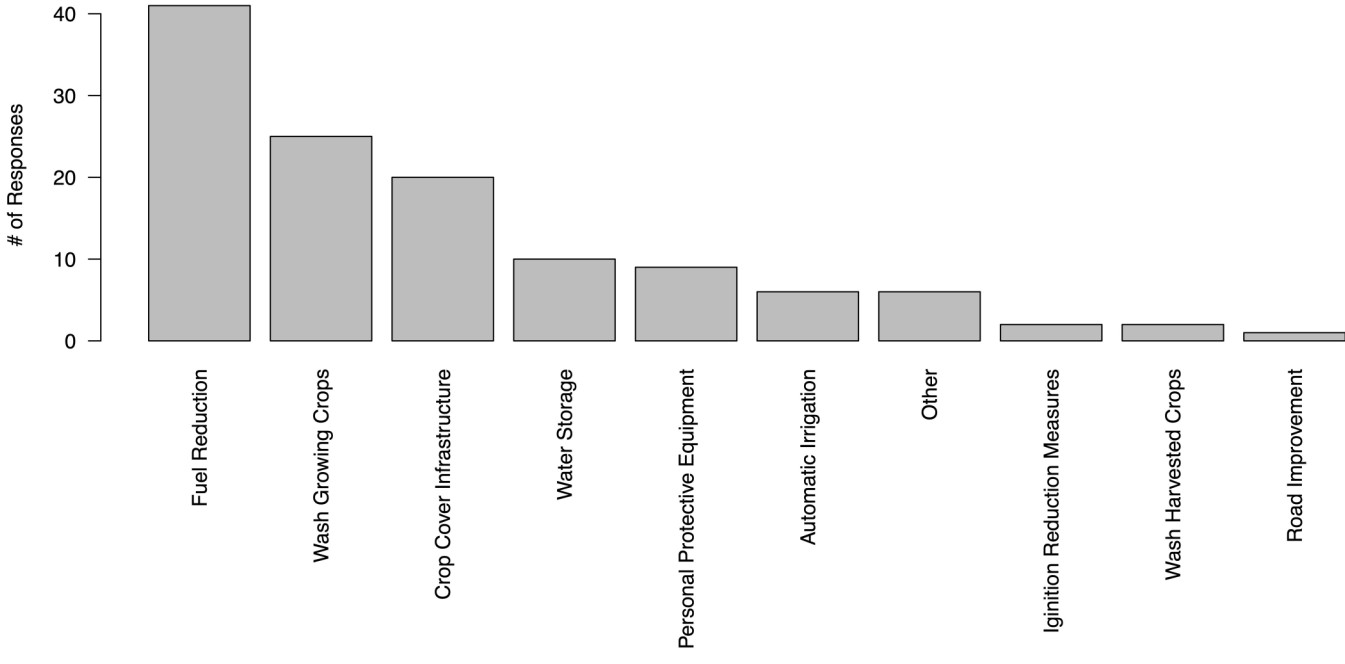

**Fig 4. Perceived effectiveness of mitigation strategies.**

reported (Fig 3) and viewed as effective (68% of respondents), water storage – also widely reported – was not viewed nearly as effective (17%). Some of this may be an artifact of our question design – one respondent noted that it was "hard to designate one item… water storage and fuel reduction are both valuable." Water storage may have also been viewed as less effective due to access issues, as noted above – if no one was at the site to use stored water following evacuation, it would not be effective at reducing fire risk – as well as voiced concerns over regulatory limits on particular forms of storage and water availability (see also [62]). Washing growing crops was viewed as most effective by 47% of respondents, followed again by crop cover infrastructure at 38%. One respondent noted the value of "[c]overing your crop, greenhouses… Keeping the smoke/ash off mitigates all but loss of light."

Other mitigation techniques were not as widely reported (Fig 3), but in both conversations with producers (as well as in narrative elaborations under our "other" category) were described as effective. These included the widely discussed use of leaf blowers to remove ash and particulates from crops, which some described as superior to washing. Additionally, while only a few (4%) reported post-harvested treatment of crops with a chemical solution, this practice – spraying or dipping with diluted hydrogen peroxide to reduce smoke taint as well as mildew/microbial impacts – was also recommended by some producers (4%).

Respondents reported a range of spending on mitigation – with a median value of $10,000 and a maximum of $275,000. Based on conversations with growers, we speculate that these more costly expenditures included road improvements, installation of water storage systems, and/or fuel reduction – in other words, large-scale infrastructure investments and land management efforts. One respondent confirmed that they spent "at least $20,000" on wildfire prevention, while another claiming "well north of $400k" pointed out that "[l]ots of overhead expenditures can cover multiple

purposes[,] i.e., [.,] water tanks can cover irrigation and fire suppression." Still others noted investments in solar power as a means of contending with power outages. (As above, given the limitations of our data – self-reported, and neither tied to year nor broken down by mitigation type – we cannot ascribe typical costs to any one mitigation strategy, nor confirm whether investments were made following (i.e., in response to) fire/smoke impacts.)

Beyond on-farm practices, producers may also look to institutional support mechanisms to help mitigate their risks, including insurance. We asked producers if they held any sort of fire insurance, its associated costs, and about their decision to not hold insurance if they did not. Only 19% of our respondents reported that they held fire insurance. For those who did not, some had "never even considered it" and others "do not like participating in the insurance world," but the majority contended either with prohibitive costs (64%) or availability limitations in their area (52%). In both our survey and earlier conversations with producers, reference was made to limitations on both crop and fire insurance related to federal status (e.g., "There's still a conflict between cannabis and legitimate insurance. We have to be careful not to get [our insurance] cancelled because we grow cannabis"), as well as the increasing costs and unavailability of fire insurance in California generally ("We tried to get fire insurance for the farm property and crop but can't find any and now can't afford it"). Indeed, since our survey California has seen multiple insurance companies rescind provision of home insurance due to increasing wildfire frequency and intensity in the state [63].

Finally, we asked producers to share their top three sources of concern or insecurity for their operations (Fig 5). Their responses highlight that while wildfire is a major issue among outdoor producers (among the top five concerns), it is often overshadowed by political economic factors – namely, prices (63%), taxes (61%), and both state and county regulatory policies (37% each, respectively). These patterns match anecdotal evidence from conversations with producers in pre-survey focus groups, webinars, and other interactions that have emphasized concerns over regulations and market fluctuations (see also [20,64]), and serve to highlight the situatedness of risk perception among producers.

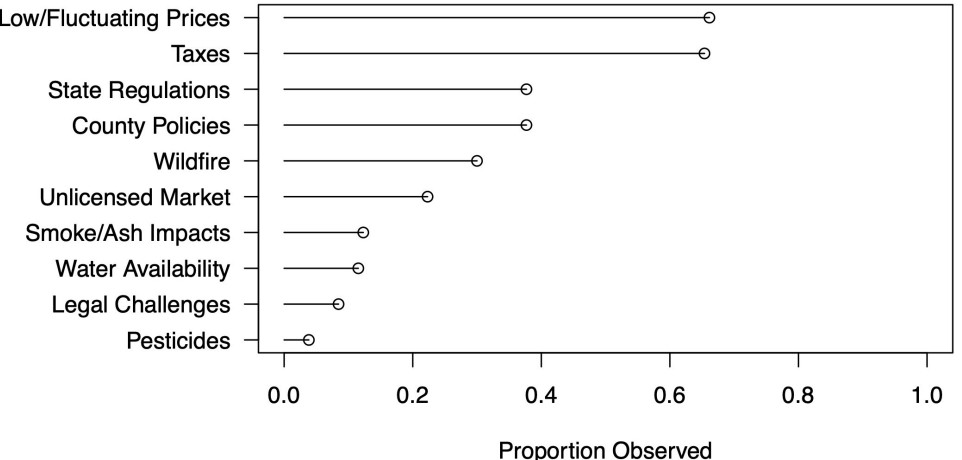

**Fig 5. Producers' top concerns.**

## Discussion

Our findings provide a preliminary overview of the diverse array of impacts wildfire has on outdoor cannabis production, as well as the range of adaptive responses producers have taken to mitigate these threats. Respondents expressed concerns over the direct and indirect effects of fire and smoke on crops themselves as well as on broader production processes. While some producers reported experiences with direct burning, others emphasized fire proximity given the broader geographical impacts of smoke plumes: "we are aware that if there is a large wildfire nearby, our operations will be very heavily affected" (see also [8]).

Wildfire's effects range from human health impacts to slowed plant growth, evacuation and loss of site access, power outages, as well as increased risk of pests and disease (Fig 1), all of which can occur far beyond wildfire perimeters. Health impacts, in particular, extend to agricultural workers broadly [65]. Respondents noted how "unhealthy air quality" and "physical stress from smoke" also hindered normal care and maintenance of crops (see our speculations above regarding the perceived (in)effectiveness of water storage). Air quality concerns corroborate findings from Beckman, et al. [14], who also noted wildfire smoke among their list of cannabis workers' physiological exposures.

To limit the direct risk of wildfire damage, producers turn to fuel reduction and other measures to prevent fires on their farms or to fight fires that do break out. These include maintaining defensible space, safety measures to reduce risk of ignition, water storage on site, and facilitating road access for firefighters. Respondents also pointed out the value of local knowledge: some related direct firsthand firefighting experience on local crews, while others drew on decades-long relationships in place to assist firefighters with site access during burns.

Addressing the impacts of smoke and ash appears to present a more difficult challenge. Some suggested dipping or washing harvested crops in a hydrogen peroxide solution, while others dried crops "in a closed environment with zero outside air coming in" to remove smoke taint. Future research might explore comparisons with smoke taint mitigation in other crops, including wine [60,66]. As noted above, there appears to be potential for processing tainted flower into saleable oil, but market and regulatory dynamics may hinder its widespread use. Finally, in contending with ash fall, producers often turned to crop cover infrastructure (e.g., hoop houses) as well as washing or mechanically blowing growing crops. Some used the previously mentioned post-harvest dipping, while others experimented with new technologies (e.g., "wind tracking software").

## Fire in context

While there is clearly a range of strategies in use by outdoor cannabis producers around the state, our research cannot speak to the effectiveness of these tools and techniques (beyond producer perceptions) for reducing the threat or impacts of fire and smoke. Indeed, some respondents communicated a sense of futility and frustration in the face of wildfire, noting that "nothing helps" and resorting to "praying" in the face of fire. Understanding this range of attitudes requires that we consider the varied impacts experienced by producers, as well as the ways in which wildfire is experienced within a broader socio-environmental context.

As shown in Fig 5, concerns over wildfire and smoke are often outweighed by concerns over other pressures on production and profitability. Several respondents directly compared wildfire with these other concerns: "wildfires have been hard but the [California] tax structure is our biggest obstacle to success"; "Aside from wildfire, the number [one] risk to cannabis business is poorly thought out regulations and tone deaf regulators"; "More than any factors, including fire, overregulation on EVERY level is devastating the industry." Indeed, for some fire was not even viewed as an issue: "The issue for outdoor/mixed light growers is not fire

concerns. [I]t's excessive regulations, punitive/excessive taxation, massive imbalance in licensing, ignoring [hierarchy] of and existing laws, and painstakingly confusing and punitive system corrupt, expensive and arduous designed intentionally to fail"; for others fire was simply overshadowed by other pressures: "The fires were rough, and more so for those more directly affected. But the price drop and oversupply, no on farm sales allowed, coupled with tax burden has allowed the black market to thrive, and understandably so. The costs of compliance are too high to prevent consumers from getting much cheaper cannabis the old fashion way" (on illicit production following legalization, see [49,67]).

## The co-production of hazard

Cannabis's ongoing status as a Schedule I controlled substance under the federal Controlled Substances Act (21 U.S.C. §801 et seq.) prevents California's licensed producers from using many of the institutions and resources available to other agriculturalists, including federal crop insurance as well as much of the formal financial system [68]. Fire insurance remains a broader issue within California, as discussed above – notably, several respondents remarked on how cannabis production on-site could also make finding homeowners insurance difficult. For cannabis producers, California's regulatory treatment vis-à-vis other forms of agriculture has created problems of site access following evacuation and road closures (Fig 1), although efforts are ongoing to bring cannabis in line with other agricultural exemptions [20,69]. Notably also, state- and county-level regulations around defensible space and water storage, as well as limitations on treatments for those proximate to public lands, have their own management implications. These create an uneven regulatory landscape that may affect adaptation and adoption of any given mitigation strategy (Fig 3), and further points to the co-production of hazards and vulnerability [19,70,71].

Many producers expressed a desire to be treated as an agricultural crop, at least insofar as this would mean greater access to resources and support (see also [20]). As one respondent put it, "Cannabis needs to be regulated like the agricultural crop that it is, and have access to insurance, grants, and research that all other crops do." Others emphasized the specific regulations surrounding the industry: "Wildfire risks, drought, etc, are common risks in agriculture. County policy, fees and taxes, and lack of retail throughout the state cause artificial difficulties on what is in reality an agricultural crop." Many of these regulations were described as exceptional and burdensome, even perceived as limiting farm sustainability and restricting the flexibility needed for adaptation to wildfire (see [19], pp. 1411–12 on the interplay between such political and material vulnerabilities).

In discussing economic pressures (e.g., fluctuating prices; Fig 5), respondents also highlighted questions of scale – particularly post-legalization dynamics of "large scale cultivation operations… driving the small farm out of business." The economic geography of legalized cannabis, with bigger farms "down south and in central California," was perceived to have "driven prices down" (see [11]). Some pointed to a similarly uneven application of environmental regulations, particularly around water consumption. As one respondent explained, "The main pressure[s] that most farmers I know are feeling are the combination of low wholesale prices, over-saturation due to increases in production from large, [corporate] financed farms, combined with the insane tax structure that is currently in place" (see [49,50]).

Outdoor and mixed-light cannabis farms are disproportionately legacy producers and smaller scale, and there may be a correlation between those contending with market pressures and those located in areas contending with higher wildfire risk [13]. There is the potential for impacts from wildfire and smoke to thus reinforce existing trends towards concentration and centralization within the industry. Concerns over wildfire thus become linked with concerns over the long-term viability of small-scale production, and fears of being pushed out

of business and often off the land through a combination of market and regulatory pressures [50]. Future research might build from these political ecology-informed concerns to explore the impacts of scale on adaptive capacity and wildfire impacts.

Producers also appear to be more or less vulnerable to fire based on the particular location of their farm. As one respondent explained, "I have a large piece of ag[ricultural] land in the floor of the valley… We have no structures so I do not retain insurance… We are also relatively safe on the valley floor – fewer trees and abundant water." In focus group conversations, producers also noted the role of topography, proximity to water, and other environmental features in both their productivity and relative vulnerability to fire and smoke. While these qualities are not adaptive responses to the threat of wildfire, they do point to the ways location, landscape features, and differing production patterns might affect resilience. Many legacy producers in rural northern counties work and live in the WUI and near public lands, which could also complicate fire prevention strategies. Multiple respondents specifically called out the U.S. Forest Service's land management practices, demonstrating possible linkages between wildfire-related tensions and longstanding anti-governmental attitudes (compare [37]).

While relocation represents a conceivable strategy for avoiding the worst fire dangers and smoke effects, few respondents saw fire or smoke affecting their choice of grow location (Fig 2). Even fewer respondents had fire/smoke motivating a shift toward indoor production – this despite comments regarding the effectiveness of "indooring" for reducing smoke effects (as one respondent noted, "The best thing growers can do to reduce smoke impact on cannabis crops is to use [g]reenhouses"). We speculate that relatively low figures around changing location and moving toward indoor production may in part reflect institutional challenges associated with the uneven regulatory landscape and the hurdles of shifting from outdoor to indoor permitting (see [20]). Fixed investments in place (which cannot be easily transferred or recouped) along with the limited availability of land elsewhere in the state (particularly given county-level variation in cannabis permitting) help account for on-farm mitigation practices being the primary strategy used by most outdoor producers. At the same time, we speculate (based on conversations with producers) that relicensing hurdles – along with the additional costs (e.g., electricity) associated with indooring production – prevent this strategy from becoming more prominent. Producers that would switch must effectively start the permitting process over – a process already fraught with delays and highly criticized. Thus we see how the current regulatory regime may be hindering mobility and the reorganization of production as adaptive strategies.

Importantly, for many producers the site of production is also their home, and ties to place further complicate their ability to relocate. As one respondent explained,

"We are both second generation cannabis farmers, and would move in a minute if we didn't farm at our home for income. We have spent so much money on nonsense regulatory fees that we are always short to make fire improvements here. Farming is stressful. Fire is stressful… If the fire comes, we lose our annual income and home, [and] that's a lot of pressure during a scary time that you have no control over…. Each year is a bigger struggle, and we watch our community of farmers struggle, and our small town that has survived on cannabis growers making and generously spending money, just dwindle."

There is no doubt that geographical location affects concerns over fire and smoke relative to other pressures. One producer in Mendocino County, for instance, noted that "[w]ildfire is the least of our concerns" in the face of "redundant, cumbersome regulations at the county and state levels." Future research might help clarify at what point proximity to fire becomes primary vis-à-vis political economic demands, yet it is clear that for many the two cannot

be easily disentangled. Reduced returns (e.g., through lower prices) mean reduced resources available for investments in mitigation, while regulatory structures hinder certain modes of adaptation (e.g., relocation or transitioning to indoor production). Based on our findings, we might expect large-scale, better-resourced producers to be more able to avoid fire and smoke through a combination of greater freedom of mobility/relocation, resources to navigate the (re)permitting processes toward indoor production, and/or investment in more costly mitigation measures (also compare [72]).

Thus we see the interrelation of institutional and market failures with environmental hazards [15,70,73], and how market dynamics and the regulations surrounding cannabis production in California might undermine the adaptive capacity of producers to respond to wildfire. In the absence of dedicated support structures for adaptation, particularly among more vulnerable communities [30,74,75], there is the potential for the current regulatory regime to reproduce cannabis's geographic marginality and wildfire to reinforce extant economic trends within the industry – all to the detriment of small producers and rural communities.

## Conclusions

Wildfire impacts are only expected to worsen, presenting a major challenge for those who live and work in the expanding landscape of fire risk across California and within the broader American West region. While the federal future of cannabis remains uncertain, the industry remains an important economic sector within the state even as it continues to evolve. The cannabis-wildfire nexus serves as a valuable lens onto the complex, interrelated dynamics of rural economies, state policy, and environmental hazards, an important site for thinking about socio-environmental co-production and adaptation to our "new normal."

We considered here the direct and indirect impacts of wildfire on outdoor cannabis production in California, as well as how producers respond to the threats of wildfire and smoke. Cannabis producers experience a range of impacts beyond direct burns, including ash fall, smoke taint, human health impacts, and disruptions of production processes. There are already a range of mitigation strategies in use among outdoor producers across the state, yet there remains an opportunity for researchers to test the effectiveness of these approaches. This could in turn aid in the promotion of best practices while bringing cannabis production – often siloed given its historically illicit status – into conversation with the broader scientific literature around wildfire adaptation (e.g., [75–77]). Future research might also investigate other themes raised in our findings, including quantification of reduced light effects on crop growth rates; new forms of laboratory testing for capturing the effects of smoke and ash exposure on consumer products; how uneven positionality and production scale might play a role in the experience of and ability to respond to fire/smoke hazards; and the effects of state- and county-level regulations on producers' adaptive capacity.

Notably, it is difficult to consider these questions without an interdisciplinary approach that recognizes environmental hazards as socially co-produced – highlighting the importance of social science and critical socio-environmental theory (e.g., political ecology) for thinking on risk perception, experiences, and decision-making (see also [71,73]). Social science insights, in turn, emphasize how questions surrounding cannabis production are difficult to disentangle from broader questions of rural political economy. Our findings confirm the need to consider wildfire as a socio-environmental challenge – co-produced and contextually experienced – including the ways in which market and regulatory pressures affect producers' adaptive capabilities. Cannabis policy may yet become a model for agriculture amid socio-environmental crisis [67], and it is our hope that our insights on the experiences and concerns of producers contending with wildfire might provide the first steps toward policy interventions to better address these challenges.

## Supporting information

**S1 File. Survey Protocol.** Survey: "Wildfire Impacts on California Cannabis Production." Survey conducted via Qualtrics.
(PDF)

## Acknowledgments

The authors wish to thank all those who took the time to respond to our survey and the organizations that helped promote it. Thanks also to UC Berkeley's Cannabis Research Center and members of the Land Use Change Lab for their assistance and input, and to the two anonymous reviewers who provided insightful comments for improving the text.

The findings and conclusions in this publication are those of the authors and should not be construed to represent any official USDA or U.S. Government determination or policy.

## Author contributions

**Conceptualization:** Jeff Vance Martin, Theodore E. Grantham, Van Butsic.

**Data curation:** Christopher Dillis, Genoa Starrs, Danielle Schell.

**Formal analysis:** Christopher Dillis, Genoa Starrs.

**Funding acquisition:** Theodore E. Grantham, Van Butsic.

**Investigation:** Jeff Vance Martin, Genoa Starrs.

**Methodology:** Jeff Vance Martin.

**Project administration:** Genoa Starrs, Theodore E. Grantham, Van Butsic.

**Supervision:** Theodore E. Grantham, Van Butsic.

**Visualization:** Christopher Dillis.

**Writing – original draft:** Jeff Vance Martin.

**Writing – review & editing:** Jeff Vance Martin, Christopher Dillis, Genoa Starrs, Danielle Schell, Theodore E. Grantham, Van Butsic.

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
