## [Decision Letter · Decision Letter 0]

25 Sep 2024

PONE-D-24-18698Wildfire impacts and mitigation strategies among California cannabis producersPLOS ONE

Dear Dr. Martin,

Thank you for submitting your manuscript to PLOS ONE. After careful consideration, we feel that it has merit but does not fully meet PLOS ONE’s publication criteria as it currently stands. Therefore, we invite you to submit a revised version of the manuscript that addresses the points raised during the review process.

We look forward to receiving your revised manuscript.

Kind regards,

Vanessa Carels

Staff Editor

PLOS ONE

Journal Requirements:

2. In the online submission form, you indicated that, raw data cannot be shared publicly because of anonymity requirements associated with human subjects review. Summarized data can be made available upon request. 

3. Please upload a copy of Figure 5, to which you refer in your text on page 16 and 18. If the figure is no longer to be included as part of the submission please remove all reference to it within the text.

Reviewers' comments:

Reviewer's Responses to Questions

**Comments to the Author**

1. Is the manuscript technically sound, and do the data support the conclusions?

Reviewer #1: Yes

Reviewer #2: Partly

2. Has the statistical analysis been performed appropriately and rigorously? 

Reviewer #1: Yes

Reviewer #2: I Don't Know

3. Have the authors made all data underlying the findings in their manuscript fully available?

Reviewer #1: No

Reviewer #2: Yes

4. Is the manuscript presented in an intelligible fashion and written in standard English?

Reviewer #1: Yes

Reviewer #2: Yes

5. Review Comments to the Author

Reviewer #1: Given the overlap between areas of high wildfire hazard and high-value cannabis production, this paper adds valuable information. However, I do have some suggestions and questions that I hope you can address.

One, you state a couple of times that cannabis is an important crop. But it also seems from a couple of statements in the paper as though the highest production does not occur among outdoor growers nor in the Emerald Triangle region. Do you have a sense of how much your survey population contributes to the industry in California as a whole?

The abstract currently provides too little information. You tells us what you did, why the topic matters, and how you hope your findings could be used. But you don’t actually report what you found. Could you include a sentence or two highlighting major findings (e.g., that most growers reported impacts from fire but they don’t see them as critical in the face of other impediments to profitability)?

On p. 10 and again on p. 12, you cite data that shows wildfires are increasing in the state, but your statements imply that prior to climate change and fire suppression, the only source of fire on California landscapes was Indigenous burning. That is not correct – fire was part of the landscape before, though probably not frequently in the Emerald Triangle. Also, a third contributor to the current trend is human ignitions, largely due to WUI growth.

I encourage you to look for repetitions. For example, the research questions are presented on p. 11 and again at the bottom of p. 13. They only need to appear once.

On p. 14, you say you focused on growers with outdoor and “mixed-light” licenses. However, the footnote explaining what “mixed-light” means isn’t linked to this first reference but to a second reference on the next page. I suggest moving the footnote to that first reference. Also, please help the reader understand how mixed-light producers differ from indoor producers, since both use greenhouses. (This could go in the footnote as well.)

A response rate of 13% is not at all unexpected, given the time demands on agricultural producers and broad trends in society of not responding to surveys. Still, it means most of the eligible respondents chose not to do so. It would be helpful to provide a sentence or so describing how your respondents might have differed from the overall population, and what biases that might have introduced.

The caption for Table 1 requires a bit of elaboration. It appears that the question referred to impacts on crop yield and quality, but that’s not obvious. (For example, in the figure you referenced at the start of Results, impacts also included human health and incidental effects of fire suppression.)

In the caption for Figure 3 and also the discussion on p. 20, you describe strategies “observed,” but these can’t actually be observed by a survey. A better word is “reported.”

On p. 23 you refer to a Figure 5, but that was not included in my copy of the manuscript. Also, it is generally thought to be more straightforward to present all results in the Results section rather than waiting for the Discussion. I understand that your manuscript might flow better this way, but I suggest moving the middle paragraph of this sub-section on producers’ top concerns to the Results section.

One issue that was briefly referenced in your paper, but not specifically discussed, is water availability. Morgan et al. (2021 in Water) found that water rights concerns in the Emerald Triangle could make it more difficult for growers to get all the water they need. In your study, water storage was seen as less effective. Is it possible that availability affected that response?

Reviewer #2: Summary

This manuscript explores the experiences California cannabis growers with wildfire in terms of impact on production and responses. In analyzing survey response data from 199 cannabis producers, the main results indicate smoke appeared to have the most impacts and provided the most challenges. The manuscript also finds that growers have made changes to cultivation techniques in response to smoke and burn threats, focusing on on-farm practices, timing of cultivation, and location. The manuscript then further explores grower perceptions of effectiveness of changing strategies, finding that fuel reduction around to be perceived as the most effective to reducing risks from wildfire. They go on to discuss how other contextual factors -prices, taxes, regulatory policies- can amplify the perceptions of risk for wildfire impacts and the additional institutional factors such as insurance polices that support the construction of vulnerability to wildfire impacts.

This is an interesting paper, though has several areas in need of improvement before publication. First, there appears to be several more research questions addressed in the paper than the two simplistic ones introduced in the Introduction. I suggest the authors identify an over-arching research question that would encompass the multitude of research questions explored through the paper. Second, theory is introduced in the discussion but receives only light engagement. I suggest looking at work in political ecology that engages in discussion about constructed or co-produced vulnerability to wildfires and then to introduce some of this theory while framing the paper in the introduction. Greater engagement with theory may help to focus your paper a little bit more as it currently seems to just focus on reporting lots of descriptive statistics.

Focused Comments

Methods

Include the information about how many survey responses you received in this section.

It is unclear about where the narrative responses came from – was this from the open-ended question included in the survey or from the focus group? Did you use particular open-ended questions to help you understand the patterns that you saw in the descriptive statistics (this would seem logical)? The way it is written now makes it appear that you analyzed all of the open-ended questions together instead of grouping them by topic. The description of your process of your qualitative analysis is pretty vague and confusing. Please clarify in general.

Results

This section seems to lack focus. It runs on for several pages of descriptive statistics. It includes responses to much, much more than the two research questions you identified.

Discussion

This needs to be further developed and situated in the literature that you identify.

A paper to look into:

Simon, G.L. and Dooling, S., 2013. Flame and fortune in California: The material and political dimensions of vulnerability. Global Environmental Change, 23(6), pp.1410-1423.

Conclusion

The main finding seems to be that vulnerability is also dependent or co-produced on institutional contexts. This is not a novel finding in itself. I’d like to challenge the authors to identify how this work may push along the literatures within which they situate the study.

6. PLOS authors have the option to publish the peer review history of their article (what does this mean? ). If published, this will include your full peer review and any attached files.

**Do you want your identity to be public for this peer review?** For information about this choice, including consent withdrawal, please see our Privacy Policy .

Reviewer #1: No

Reviewer #2: No

---

## [Author Response · Author response to Decision Letter 1]

8 Jan 2025

Dear PLOS ONE Editors and anonymous reviewers,

Thank you so much for your detailed read of and for the opportunity to respond to the reviewers’ comments on our manuscript, “Wildfire impacts and mitigation strategies among California cannabis producers" -- as well as for your patience as we made the necessary changes.

Please find attached a response table to the points raised by the academic editor and reviewers. We believe we have adequately responded to each issue, but welcome an opportunity to make any additional revisions necessary.

Thank you again for your consideration!

---

## [Decision Letter · Decision Letter 1]

7 Mar 2025

Wildfire impacts and mitigation strategies among California cannabis producers

PONE-D-24-18698R1

Dear Dr. Martin,

We’re pleased to inform you that your manuscript has been judged scientifically suitable for publication and will be formally accepted for publication once it meets all outstanding technical requirements.

Kind regards,

Asghar Khan

Academic Editor

PLOS ONE

Additional Editor Comments (optional):

Dear author,

We appreciate the effort you have put into revising the manuscript and addressing our concerns. The revisions have strengthened the study, and the additional theoretical insights from political ecology provide valuable context. The manuscript is now well-structured, technically sound, and the conclusions are well-supported by the data presented.

One remaining issue is data availability. It is unclear whether any de-identified data will be made publicly accessible. While we understand that institutional restrictions (e.g., from UC Berkeley) may limit this, providing as much data as possible would enhance transparency and reproducibility. If restrictions prevent full data sharing, a clear justification in the manuscript would be beneficial.

Overall, the study makes an important contribution, and we have no further major concerns. Thank you for your thoughtful revisions.

Reviewers' comments:

Reviewer's Responses to Questions

**Comments to the Author**

1. If the authors have adequately addressed your comments raised in a previous round of review and you feel that this manuscript is now acceptable for publication, you may indicate that here to bypass the “Comments to the Author” section, enter your conflict of interest statement in the “Confidential to Editor” section, and submit your "Accept" recommendation.

Reviewer #1: All comments have been addressed

Reviewer #3: (No Response)

2. Is the manuscript technically sound, and do the data support the conclusions?

Reviewer #1: Yes

Reviewer #3: Yes

3. Has the statistical analysis been performed appropriately and rigorously? 

Reviewer #1: N/A

Reviewer #3: Yes

4. Have the authors made all data underlying the findings in their manuscript fully available?

Reviewer #1: No

Reviewer #3: Yes

5. Is the manuscript presented in an intelligible fashion and written in standard English?

Reviewer #1: Yes

Reviewer #3: Yes

6. Review Comments to the Author

Reviewer #1: I appreciate your attention to the suggestions I made on the original submission. I'm satisfied that you addressed them all satisfactorily. Also I appreciated the additional theoretical insights from political ecology.

Reviewer #3: An excellent and tidy study of the impacts of wildfire on cannabis cultivators. Although, I do not see the fire risk lessening any time soon, and the life cycle of cannabis has its peak during fire season. I do not know what the long term solutions are for these farmers, other than changing to indoor production or moving their farm to a less fire prone area, which, in CA doesn't really exist anymore.

7. PLOS authors have the option to publish the peer review history of their article (what does this mean? ). If published, this will include your full peer review and any attached files.

**Do you want your identity to be public for this peer review?** For information about this choice, including consent withdrawal, please see our Privacy Policy .

Reviewer #1: No

Reviewer #3: No

---

## [Editor Report · Acceptance letter]

PONE-D-24-18698R1

PLOS ONE

Dear Dr. Martin,

I'm pleased to inform you that your manuscript has been deemed suitable for publication in PLOS ONE. Congratulations! Your manuscript is now being handed over to our production team.

Kind regards,

on behalf of

Dr. Asghar Khan

Academic Editor

PLOS ONE